# Sequence Alignment-based Similarity Metric in Evolutionary Neural Architecture Search

Mateo Ávila Pava[1]  René Groh[1]  Andreas M. Kist[1]

[1]Department of Artificial Intelligence in Biomedical Engineering,
Friedrich-Alexander-Universität Erlangen-Nürnberg (FAU), Erlangen, Germany

**Abstract**  Neural Architecture Search (NAS) has emerged as a powerful method for automating the design of deep neural networks across diverse applications, with evolutionary optimization showing particular promise in addressing its intricate demands. However, the effectiveness of this approach highly depends on balancing exploration and exploitation, ensuring that the search does not prematurely converge to suboptimal solutions while still achieving near-optimal outcomes. This paper addresses this challenge by proposing a novel similarity metric inspired by global sequence alignment from biology. Unlike most of the existing methods that require trained model weights for comparison, our metric operates directly on neural network architectures within the defined search space, eliminating the need for model training when comparing two architectures. We outline the computation of the normalized similarity metric and demonstrate its application in quantifying diversity within populations in evolutionary NAS. Experimental results conducted on popular datasets for image classification, such as CIFAR-10, CIFAR-100, and ImageNet16-120, show the effectiveness of our approach in guiding diversity based on our suggested control function. Additionally, we highlight the usefulness of our similarity metric in comparing individuals to identify advantageous or disadvantageous architectural design choices. The code is available at https://github.com/ankilab/evonas_similarity_metric.

## 1 Introduction

The field of artificial intelligence (AI) has recently made significant progress, primarily due to the increasing sophistication of neural networks. With the rapid emergence of diverse architectural design choices (Szegedy et al., 2015; He et al., 2016; Chollet, 2017), the process of crafting neural network architectures is becoming more and more complex. This complexity has made manual design approaches less practical, emphasizing the need for automated methods, commonly referred to as Neural Architecture Search (NAS). NAS involves exploring a vast space of possible network structures and optimizing for performance metrics such as accuracy, efficiency, and computational cost to ultimately select the most effective architecture for a given task or dataset (Elsken et al., 2019). Besides reinforcement learning (Tan et al., 2019; Zoph and Le, 2016), bayesian optimization (Jin et al., 2019; Kandasamy et al., 2018; White et al., 2021) or gradient-based optimization (Liu et al., 2018), genetic optimization through evolutionary algorithms (EAs) is also commonly used for NAS.

This meta-heuristic is inspired by natural selection, uses principles like survival of the fittest, and iteratively improves solutions to complex problems, such as NAS (Simon, 2013; Liu et al., 2021). However, one huge limitation of evolutionary NAS algorithms is the constant trade-off between exploration and exploitation when optimizing a neural architecture for a given problem (Tan et al., 2009). In the following, we refer to an individual as an individual in a population in terms of genetic optimization, which reflects in this case a single neural network architecture.

In this work, we present a novel neural architecture similarity measure inspired by the sequence alignment of biological sequences, which we use to control diversity within evolutionary NAS (see Figure 1). Our main contributions are:

- A novel similarity measure facilitating the comparison of candidate neural network architectures sampled from a search space in the context of NAS.

- Demonstrating the ability of the similarity measure to balance between exploitation and exploration.

- We show that our similarity measure can effectively identify unfavorable design choices by comparing pairs of individuals with high similarity but substantial differences in fitness. This allows gaining knowledge of neural network architectural design or refining the search space.

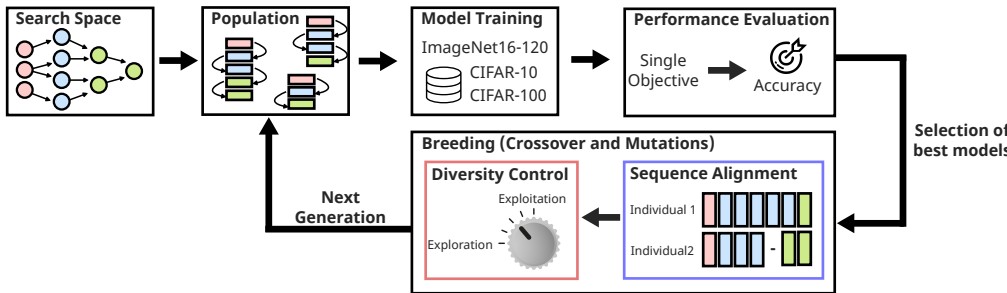

Figure 1: Overview of this study.

## 2 Background

### 2.1 Similarity of Neural Network Models

Understanding and quantifying the similarity of neural networks presents a complex challenge due to the diverse perspectives on their comparability. A recent survey categorizes nearly 50 measures (Klabunde et al., 2023) into Functional Similarity Measures (FSM) and Representational Similarity Measures (RSM). FSM compares networks based on output behaviors, such as ModelDiff (Shah et al., 2023) and Disagreement (Madani et al., 2004). RSM, including PWCCA (Morcos et al., 2018) and CKA (Kornblith et al., 2019), analyzes activations across layers. Additionally, it includes metrics like Aligned Cosine Similarity (Hamilton et al., 2016) and Geometry Score (Khrulkov and Oseledets, 2018). Both FSM and RSM are essential, as similar outputs may stem from different representations, and significant dissimilarity in output should be reflected in RSM. Choosing the most appropriate similarity measure, one that aligns closely with the specific requirements of your problem is crucial for accurate assessment.

### 2.2 Diversity in Evolutionary Algorithms

The effectiveness of evolutionary algorithms (EAs) can be affected by two main factors: (i) the loss of diversity in early generations, which results in limited exploration, and (ii) high mutation rates, which can undermine the effectiveness of the recombination operator by introducing excessive randomness and hindering the convergence of solutions (Bhattacharya, 2014). To address these challenges by directly or indirectly controlling diversity, various strategies have been proposed. These include multinational evolutionary algorithms (Ursem, 1999), crowding techniques (Mengshoel and Goldberg, 2008), mass extinction models (Krink and Thomsen, 2001), and Dynamic convergence−diversity guided EA (Li et al., 2018), among others. By providing guidance and exerting greater control over diversity, EAs can balance exploration and exploitation, improving their ability to navigate complex problems and achieve reliable outcomes.

## 2.3 Sequence alignment in biology

DNA, RNA, and protein sequences encode specific downstream functionalities (Wilson et al., 2018). For example, a natural protein sequence consists of a combination of the 20 canonical amino acids (e.g., glycine, serine, ...). The main assumption is that proteins with similar sequences have similar functions. This means that by comparing and aligning two protein sequences, one can infer evolutionary conserved regions or functional motifs. Similarly, two or more DNA and RNA sequences can be aligned with each other, which is important for the arrangement of next-generation sequencing data (Li and Homer, 2010). A well-known algorithmic solution is the Clustal family (Chenna et al., 2003; Sievers et al., 2011), which also serves as the inspiration for this work.

## 3 Proposed similarity metric based on sequence alignment

Global sequence alignment, exemplified by the Needleman-Wunsch (NW) algorithm (Needleman and Wunsch, 1970), systematically compares entire sequences like DNA, RNA, or proteins to reveal evolutionary relationships and conserved regions. This dynamic programming technique evaluates all possible alignments, assigning scores to matches, mismatches, and gaps, ultimately identifying the alignment with maximum similarity.

In this light, we propose employing sequence alignment to compare individual sequences representing neural networks within the EA's population. By applying this approach globally to entire populations, we aim to identify and uncover effective design patterns, while also gaining insights into the evolution of population diversity across generations. To the best of our knowledge, this is the first time that sequence alignment is used to compare the similarity between neural networks.

## 3.1 Search space

While distinct from Cell-based approaches like NASNet (Zoph et al., 2018) or DARTS (Liu et al., 2018), our search space falls within the Chain-Structured category (White et al., 2023). It offers flexibility for easy modifications due to its sequential architectural topology. Comprising primitive layers and hand-crafted blocks such as BottleneckBlock and ResidualBlock from ResNets (He et al., 2016), our search space includes feature extraction (F) and fully connected dense (D) layers. Each neural network sample consists of 1-20 F layers chosen from 9 predefined options, and 0-2 D layers, including Dropout and Dense layers. A randomly selected Global pooling layer, chosen among Flatten, Global Average Pooling, and Global Max Pooling, facilitates feature aggregation between the F and D blocks. The macrostructure of individuals and available layers for each block, along with their parameters, is depicted in Figure 2.

## 3.2 Model representation as sequence

To employ sequence alignment, our models must first be represented as sequences, with each character corresponding to a distinct layer or block in our search space. Each block or layer is assigned a unique character, as detailed in Table 1. See Figure 3 for the conversion of network model architecture to a sequence.

| Conv2D | DepthwiseConv | MaxPooling | AveragePooling | GlobalAveragePooling | GlobalMaxPooling | ReLU |
|--------|---------------|------------|----------------|----------------------|------------------|------|
| C | D | M | A | g | G | L |
| **BatchNorm** | **InstanceNorm** | **Dropout** | **Dense** | **ResidualBlock** | **BottlenetBlock** | **Flatten** |
| B | I | O | F | R | T | E |

Table 1: Representation of characters employed in the translation of chromosomes into sequences.

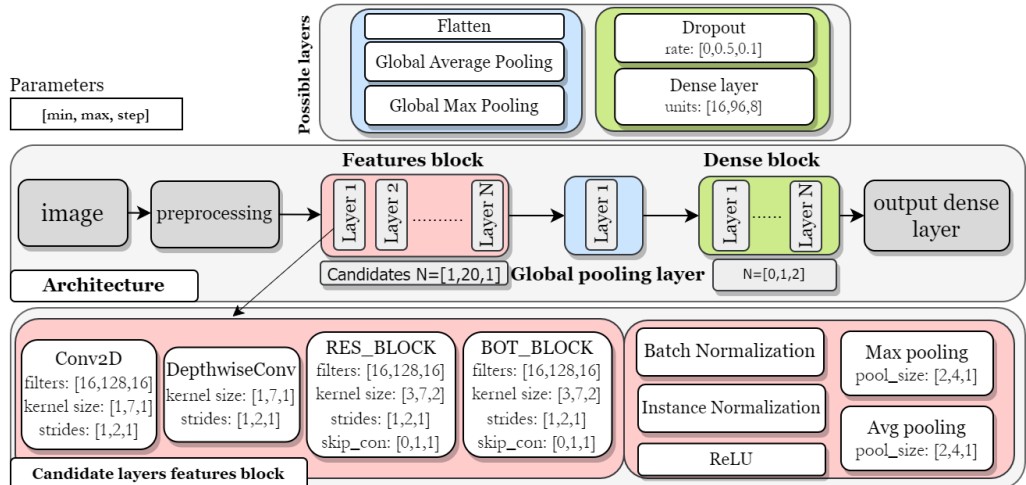

Figure 2: **Center**: the main blocks of each architecture candidate. **Top**: Possible layers for the Global Pooling block (blue) and Dense block (green). **Bottom**: Layer options for the Features block with the restrictions that the initial layer selection must be made exclusively from the choices available in the **Bottom-left** container.

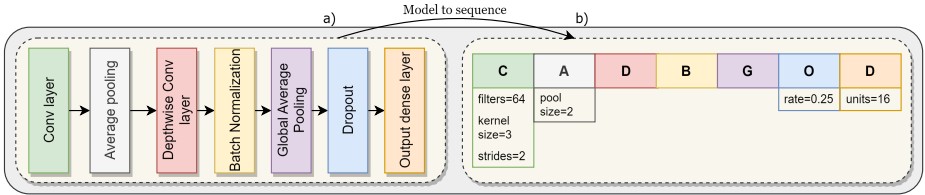

Figure 3: Conversion from neural network model architecture to a sequence representation. a) Illustration of the neural network architecture. b) Schematic representation of the network as a sequence, highlighting the parameters used for comparing one sequence to another one.

### 3.3 Sequence matching functions and gap penalties

Neural networks, being more complex than simple sequences, require consideration of layer-specific properties in similarity measures. Unlike traditional sequence alignment methods, we propose a customized matching function that accounts for differences in layer types and associated parameters.

In our matching-mismatching function, we evaluate layer similarity based on parameters. For example, comparing a Conv2D layer (C) with a ReLU layer (L) results in a mismatch value of -2, indicating incomparability. However, comparing a Conv2D layer (C) with a DepthwiseConv2D layer (D) allows for assigning a value reflecting their operational similarity, ranging from 0 to 2. We establish comparison criteria for each layer type, weighting parameters accordingly. This approach facilitates penalization or prioritization of alignment, accommodating specific block alignments (e.g., ResBlock, BottlenetBlock) over others (e.g., Dropout, InstanceNormalization). The penalty for introducing gaps or extending a sequence remains constant at -0.5.

### 3.4 Population Similarity Matrix

Custom matching functions were defined for the layers in our gene pool, enabling us to determine the absolute global alignment score between pairs of individuals. However, direct comparisons between individuals with different chromosome sizes are hindered by the absolute nature of the alignment scores, which are influenced by the number of genes in the sequences. To address this,

| MatchScore$(X, Y, \alpha, \beta) = \alpha \dfrac{\left\|X_\beta - Y_\beta\right\|}{max(\beta) - min(\beta)}$ | | |
|---|---|---|
| **if Gene $G_1$ == C && Gene $G_2$ == C** | **Weight** | **Max match score: 3** |
| $m$ = MatchScore$(G_1, G_2, \alpha_f, f)$ | $\alpha_f = 0.6$ | f: Number of filters of Conv2D
max(f) and min(f) from gene pool |
| $m+$ = MatchScore$(G_1, G_2, \alpha_k, k)$ | $\alpha_k = 0.8$ | k: kernel size
max(k) and min(k) from gene pool |
| $m+$ = MatchScore$(G_1, G_2, \alpha_s, s)$ | $\alpha_s = 0.6$ | s: Stride
max(s) and min(s) from gene pool |
| score = $3 - m$ | | |
| **if Gene $G_1$ == C && Gene $G_2$ == D** | | **Max match score: 2** |
| $m$ = MatchScore$(G_1, G_2, \alpha_k, k)$ | $\alpha_k = 1.2$ | max(k) and min(k) for both layers from gene pool |
| $m+$ = MatchScore$(G_1, G_2, \alpha_s, s)$ | $\alpha_s = 0.8$ | max(s) and min(s) for both layers from gene pool |
| score = $2 - m$ | | |
| **else if Gene $G_1$ == C && Gene $G_2$! = C** | | **Mismatch score: -2** |
| **Open gap penalty: -0.5** | | |
| **Extend gap penalty: -0.5** | | |

Table 2: Illustration depicting the definition of matching functions for Conv(C) and DepthwiseConv(D) layers, determining the alignment of genes $G_1$ and $G_2$. These functions are established for all possible layers and subsequently incorporated into the global sequence alignment process.

we normalize the global alignment to obtain a relative alignment score between sequences $S_1$ and $S_2$. A score of 1 signifies identical sequences, while a score of 0 indicates complete dissimilarity or no possible alignment.

The normalization process is based on the maximum possible alignment value, which is determined by aligning $S_1$ with itself and $S_2$ with itself. The lowest alignment score occurs when no elements align and gaps equal to the sequence length are introduced. With our gap penalty of 0.5 being lower than the mismatch penalty of 2, the lowest alignment score is given by the number of introduced gaps multiplied by -0.5. Equation 1 shows the computation of this similarity, and illustrative calculations are provided in Table 3. The penalty for introducing gaps or extending a sequence is consistently set at -0.5. This relatively low value, compared to the mismatch penalty, allows for increased flexibility in aligning larger models with smaller ones, facilitating the discovery of potential local alignments.

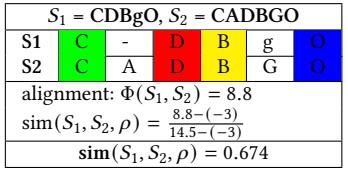

$$\Phi_{\max}(S_1, S_2) = \max\left(\Phi(S_1, S_1), \Phi(S_2, S_2)\right)$$

$$\Phi_{\min}(S_1, S_2) = \rho \cdot \max\left(\text{len}(S_1), \text{len}(S_2)\right)$$

$$\text{sim}(S_1, S_2, \rho) = \frac{\Phi - \Phi_{\min}}{\Phi_{\max} - \Phi_{\min}}$$

(1)

Figure 4: Computation of normalized similarity score between sequences $S_1$ and $S_2$, derived from the pairwise global sequence alignment score $\Phi(S_1, S_2)$ and equation 1.

Leveraging the relative similarity formula, we can numerically compare all individuals within the population even before they are trained. This process generates a square similarity matrix of size $N \times N$, where N represents the population size. The diagonal of this matrix corresponds to the self-alignment of each element, resulting in values equal to 1. This similarity matrix serves as a valuable tool for monitoring and understanding the evolution of the population over generations. Additionally, it enables the identification of design patterns, clusters of similar individuals, and potential novel architectures. This information aids in diagnosing errors that may arise during the algorithm's execution, such as premature convergence or gene convergence. Figure 5 illustrates

similarity measure matrices for an exemplary EA run on the ImageNet16-120 dataset. Individuals are sorted by model size in MB.

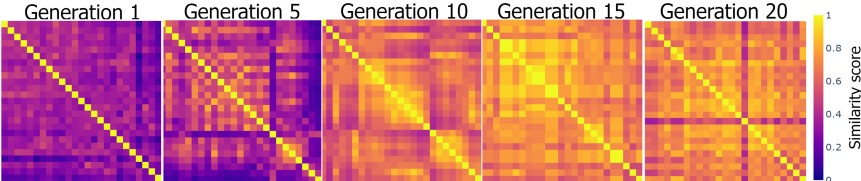

Figure 5: Population similarity matrix arranged by individual size across five generations during the execution of evolutionary neural architecture search with a population size of 25 on ImageNet16-120.

## 4 Diversity control in Evolutionary NAS

The similarity matrix allows assessing the diversity within a population and tracking its evolution across generations. A straightforward approach involves calculating the median of all off-diagonal elements, providing a global measure of population diversity ($s_{gen}$) for each generation $gen$, Where a value of 1 denotes no diversity and a value of 0 signifies maximum diversity. The evolution of diversity is influenced by factors beyond the search space, including the composition of the initial population, the choice of the crossover operator, and the mutation rate. We hypothesize that incorporating diversity guidance boosts EA achieved optimal fitness and accelerates convergence, eliminating the necessity for extensive tuning of all operators and parameters while minimizing resource consumption or working with limited population sizes.

### 4.1 Diversity control algorithm

To achieve a balanced Exploration-Exploitation equilibrium, we present a population diversity control strategy utilizing our similarity metric. This strategy iteratively generates populations using standard operators until the median similarity difference ($\Delta s$) between generations falls within the limits set by our control function. This mechanism seamlessly guides the breeding process within the evolutionary algorithm, ensuring convergence without requiring extensive parameter tuning (Figure 1). The proposed "Rigid" control function for the allowed $\Delta s$ is defined mathematically (Eq. 2). Our aim is to moderate diversity reduction while avoiding stagnation or slow growth. Upon reaching half of the total generations ($\frac{N_{gen}}{2}$), we restrict abrupt fluctuations in diversity to facilitate fine-tuning of local optima and ensure algorithm convergence.

$$g(\text{gen}) = 0, \ f(\text{gen}) = \begin{cases} 0.1 - 0.2 \left( \frac{\text{gen}}{N_{\text{gen}}} \right)^2, & \text{if gen} < \frac{N_{\text{gen}}}{2} \\ 0.05, & \text{if gen} \geq \frac{N_{\text{gen}}}{2} \end{cases} \tag{2}$$

such that $0 < g(\text{gen}) < (s_{\text{gen+1}} - s_{\text{gen}}) \leq f(\text{gen})$

Initially, we conducted testing and implemented rigid control, which can be changed to accommodate other user-defined functions to guide diversity. The outlined process is encapsulated within the Breeding algorithm, as depicted in the pseudo-code presented in Algorithm 1.

## 5 Experiments and results

This paper aims to demonstrate the versatility of our similarity metric, highlighting its effectiveness in evolutionary algorithms for diversity control and in identifying patterns and clusters within datasets. We conducted an evolutionary neural architecture search across three image classification datasets: CIFAR-10, CIFAR-100, and ImageNet16-120. ImageNet16-120, a downscaled variant of

**Algorithm 1** Diversity control using Sequence Alignment-based Similarity Metric

---

**Require:** Population similarity matrix $S_i$, Max iterations to converge $\rho$, Mutation rate $\beta$, Fittest individuals current population $F_i$, Generation $i$

1: $s_i \leftarrow \text{median}(S_i)$
2: $iter \leftarrow 0$
3: **while** $max(0.1 - \frac{i^2}{2000}, 0.05) < (s_{i+1} - s_i) \leq 0 \, \& \, (iter < \rho)$ **do**
4:      $P_{i+1} \leftarrow \text{CROSSOVER}(F_i)$                          ▷ One-point. Based on Fitness proportionate selection
5:      $P_{i+1} \leftarrow \text{MUTATION}(P_{i+1}, \beta)$                        ▷ Random gene added, removed or changed
6:      $s_{i+1} \leftarrow \text{median}(\text{ALIGN\_SEQUENCES}(P_{i+1}))$           ▷ Median of similarity matrix
7:      $iter \leftarrow iter + 1$
8:      **if** iter%5 = 0 **then**
9:          $\beta \leftarrow max(\beta - 2, 0)$
10:     **end if**
11: **end while**
12: **return** $P_{i+1}$

---

ImageNet, contains images resized to 16×16 pixels and labels ranging from 1 to 120, as defined in NAS-BENCH201 (Dong and Yang, 2020). This downscaled version reduces computational costs while serving as a valuable benchmark (Chrabaszcz et al., 2017). Our experiments involved ten different seeds and utilized separate training and evaluation sets for each dataset. CIFAR-10 and CIFAR-100 comprised 50k training images and 10k test images, while ImageNet16-120 included 151k training images and 6k test images. For training each individual neural network architecture, we utilized Stochastic Gradient Descent with a Nesterov momentum rate set to 0.9. The learning rate followed a cosine decay schedule, starting from 0.1 and tapering to 0, with a weight decay parameter of 0.0005. We used a batch size of 256 and trained for 12 epochs. In our evolutionary NAS framework, we progressed through 20 generations with a population size of 25. The top 5 models from each generation were selected based on performance. We applied a mutation rate of 20% and imposed architectural constraints limiting the number of feature layers to 20 and dense layers to a maximum of 2. Furthermore, we followed established methodologies from benchmark NAS-BENCH-201 (Dong and Yang, 2020) and its extension NATS-BENCH (Dong et al., 2020) for selecting data augmentation techniques and configuring hyperparameters. All searches were conducted using an NVIDIA A100 GPU, and due to resource constraints inherent in NAS algorithms, we limited our search to 25 individuals per generation.

### 5.1 Fitness and diversity accross generations

Two sets of experiments were conducted for each dataset to assess the impact of our diversity control strategy: one without any control (referred to as "Naive") and another incorporating our rigid control function ("Diversity Control"). The combined results for ImageNet16-120 across all seeds are depicted in Figure 6, presenting maximum fitness scores and diversity values across generations. Additional results for CIFAR-10 and CIFAR-100 can be found in Appendix B, while results for individual seeds are detailed in Appendix C. For CIFAR-100, diversity control tests achieved a 0.6% higher fitness score compared to the Naive tests. Notably, diversity control led to higher fitness scores in most generations, except for generations 8 and 17. Similarly, for ImageNet16-120, a 1% improvement was observed with diversity control, and the best fitness scores occurred between generations 10 and 20. For CIFAR-10, only a 0.4% improvement was achieved with diversity control compared to the Naive method.

Our diversity control function impacted all datasets, facilitating algorithm convergence and focusing on the best individuals near local optima during exploitation. For ImageNet16-120, diversity peaked at 0.75, while for CIFAR-100 and CIFAR-10, it reached 0.68 and 0.69, respectively. In contrast, without diversity control, diversity fluctuated between 0.55 and 0.64 for both datasets from the fifth generation onwards. Notably, with diversity control, diversity exceeded 0.8 for some seeds in ImageNet16-120 and CIFAR-10, indicating excessively small differences among individuals and negatively impacting algorithm performance. Thus, our diversity function could be enhanced by

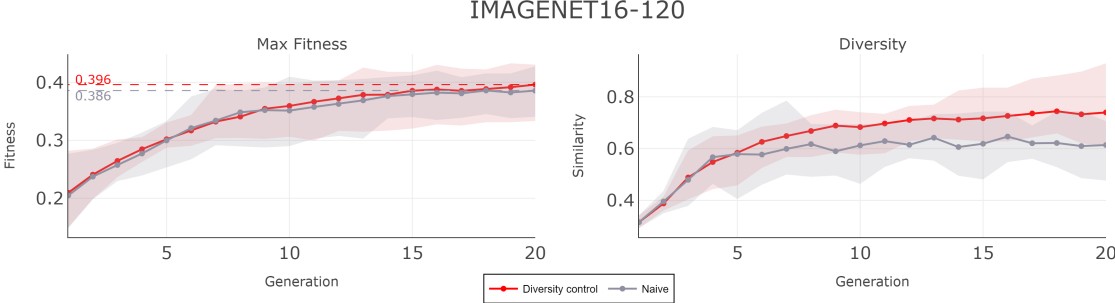

Figure 6: Evolution of Maximum Fitness Score and Population Diversity across Generations for Naive and with Diversity Control evolutionary NAS. The solid line represents the median score, while the shaded area indicates the range between the maximum and minimum values observed across 10 tested seeds.

considering both the changes in diversity between consecutive generations ($\Delta s$) and the absolute diversity value for each generation ($s_{gen}$).

## 5.2 Similarity among best individuals

Particularly significant is the comparison of models with the highest fitness for each seed, visualized as unaligned sequences in Figure 7. The Alignment chart reveals insights into gene (layer) distribution and prevailing consensus for each test. Notably, the BottleneckBlock with skip connection (red) and the DepthwiseConv (green) are prevalent among the best solutions, contributing to model efficiency and lightweight design. For CIFAR-100, the Naive test shows only three out of ten individuals with more than 13 genes, contrasting with eight out of ten individuals in the Diversity control test.

In ImageNet16-120, both Naive and Diversity control scenarios predominantly yield individuals with fewer than 13 genes. However, Diversity control generates individuals with a higher occurrence of R (ResidualBlock) and T (BottleneckBlock) blocks, suggesting a tendency toward exploring more complex solutions. This trend is reflected in the sizes of their models, as detailed in Appendix D.

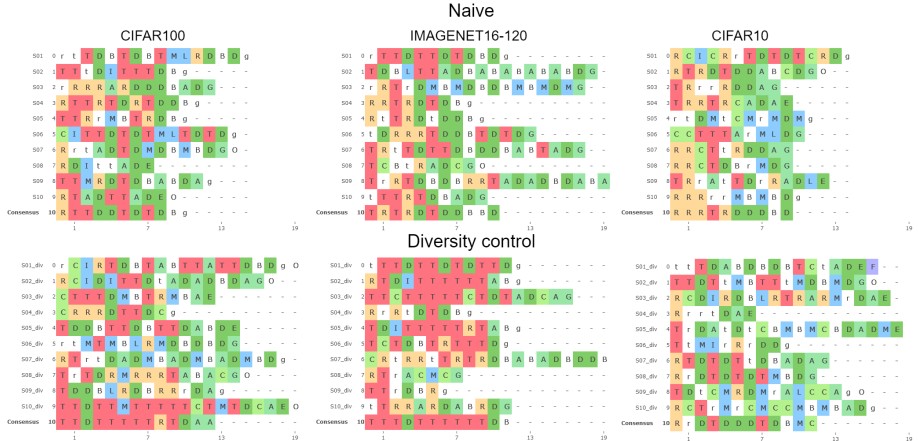

Figure 7: Alignment Chart of Sequences for Individuals with the Highest Fitness for Each Seed in Naive and Diversity Control Evolutionary NAS.

The clustergram in Figure 8 further organizes these individuals hierarchically into similarity clusters, revealing insights into convergence patterns. Notably, for CIFAR-100 and CIFAR-10

datasets, solutions from both Naive and Diversity control methods exhibit significant disparities. Most regions show similarities below 0.5, with few instances reaching 0.6 to 0.7, suggesting diverse local optima discovery. For ImageNet16-120, Naive testing reveals a clustered solution set with similarities of 0.55 to 0.65, while Diversity control identifies a larger cluster (five individuals) with similarities of 0.6 to 0.82. Models are more similar due to repeated BottleneckBlocks occurrence.

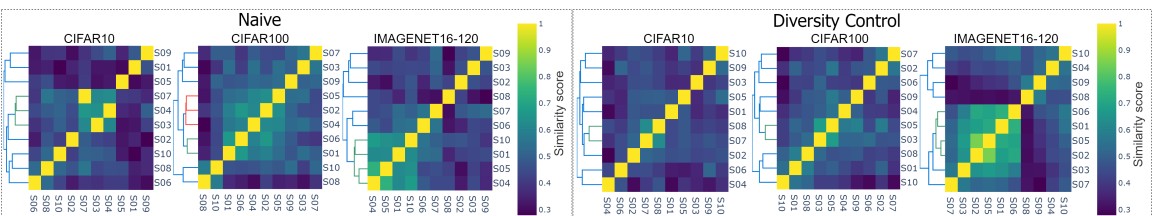

Figure 8: Hierarchical clustering based on similarity of individuals with highest fitness for each seed (S01-S10) and dataset.

## 5.3 Exploring Intriguing design patterns

Figure 9 represents the frequency of occurrence of each gene or block of 2 or more genes and was created based on the sequence representation depicted in Figure 7. Notably, the six genes with the highest number of repetitions are consistent across all datasets. In terms of blocks comprising 2 or more genes, the top three most frequently repeated blocks are TD, DB, and TT, observed across both datasets. Interestingly, CIFAR-100 exhibits a greater diversity of 2-gene combinations compared to ImageNet16-120. Additionally, the primary 3-gene blocks observed are TDB, TTD, and TDT. These findings underscore the significance of these layers in efficient image classification models such as MobileNet, ResNet, and DenseNet, among others.

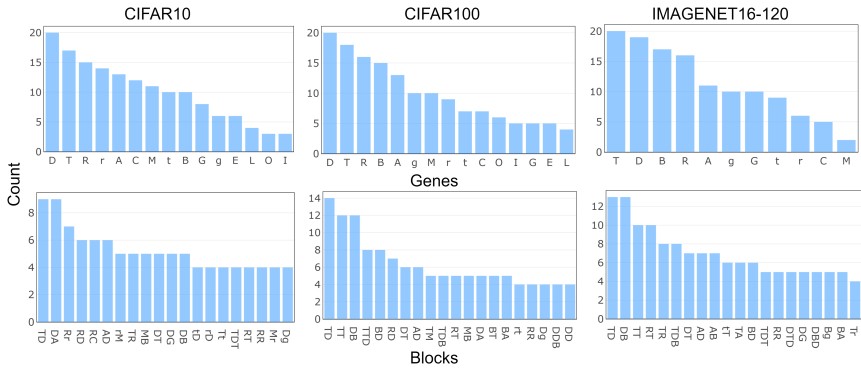

Figure 9: Frequency distribution of individual genes or gene blocks present within the top 20 individuals identified for each dataset.

An essential application of our similarity metric, particularly evident through our similarity matrix, lies in identifying outliers and instances where there exists significant similarity between individuals but considerable disparity in fitness scores. This allows us to pinpoint which genes and their respective positions contribute positively or negatively to achieving a desirable fitness score. In Appendix F, we present some intriguing design choices, showcasing all three tested datasets.

## 6 Conclusion and Future Work

In this study, we introduced a novel similarity metric based on bioinformatics sequence alignment techniques to compare neural network architectures within a defined search space. This metric

effectively measured population diversity and was integrated into an evolutionary NAS framework to balance exploration and exploitation. Our proposed rigid control function efficiently reduced diversity and exploited different local optima across datasets. The analysis of the best individuals highlighted the significant impact of specific genes, such as DepthwiseConvolution (D) and BottleneckBlock (T), on fitness scores. Moreover, we identified design cases with high similarity but notable differences in fitness scores, often involving the insertion or removal of BatchNormalization layers. Our findings are in line with recent works, where the use and selection of normalization layers were found substantial in model performance (Neubig and Kist, 2023).

However, it's crucial to acknowledge that due to computational constraints, we conducted limited experiments with 10 initial populations and the rigid control function on three datasets. Additionally, our search space doesn't provide insights into the global optimum, and evaluating all possible individuals within it is computationally expensive. Future work could involve testing with different population sizes and further exploration of different control functions as well as testing our metric in other chain-based search spaces. One limitation of our method is its limited applicability to cell-based architectures. It is better suited for chain-based or sequential architectures since our primary objective was to develop an intuitive and efficient method that facilitates straightforward comparisons among simple and sequential architectures. In future studies, the search space should also consider non-trainable layers (such as activation and pooling) to have a fixed order, as these are typically sequential but freely interchangeable, which could otherwise potentially reduce the performance of our proposed method.

As a next step, it would be valuable to extend our method to accommodate tree-like or graph-like architectures by adopting a more general approach, such as utilizing the Graph Edit Distance. Instead of solely defining matching functions as we did, we would incorporate insertion, deletion, and substitution functions. However, this would inevitably increase the complexity of the problem. Consequently, combining sequence alignment for the macrostructure or simple genes (operations) with Graph Edit Distance for the cell-based blocks commonly employed in NAS benchmarks would enhance the versatility of our method in the future.

## 7 Broader Impact Statement

Our research aims to enhance Neural Architecture Search (NAS) efficiency by improving convergence speed and optimization quality. While our method represents an advancement in the field, it requires intensive computational resources due to its exhaustive search approach. To address this, integrating our proposed similarity metric with more resource-efficient methodologies, like few-shot or single-shot NAS approaches, is a promising future direction. This aligns with our approach of comparing neural architectures without model training, offering potential sustainability benefits by reducing the environmental footprint associated with high computational demands.

**Acknowledgements**. We would like to thank Stefan Dendorfer and Lea Dang for their valuable feedback during the development phase. We receive funding from the Bavarian State Ministry of Science and the Arts (StMWK) and Fonds de recherche du Québec (FRQ) under the Collaborative Bilateral Research Program Bavaria – Québec managed by WKS at Bavarian Research Alliance (BayFOR) and Fonds de recherche du Québec – Santé (FRQS). The presented content is solely the responsibility of the authors and does not necessarily represent the official views of the above funding agencies.

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

## A  Diversity control functions

We confined our main work to a single specific control function due to the significant computational resources required to test various seeds with each function. However, different functions can be proposed depending on the desired approach to diversity control.

An alternative function to prevent excessively restricting population diversity increase is to implement a linear functions-based on the total number of generations ($N_{gen}$), ensuring that the lower bound function $g(gen)$ does not remain constant at zero throughout all generations. This method represents a softer or more flexible control strategy and is defined in equation 3.

$$
\begin{aligned}
f(\text{gen}) &= \begin{cases} 0.025, & \text{if gen} < \frac{N_{gen}}{2} \\ 0.05(\frac{\text{gen}}{N_{\text{gen}}}), & \text{if gen} \geq \frac{N_{gen}}{2} \end{cases} \\
g(\text{gen}) &= -0.1 + 0.1(\frac{\text{gen}}{N_{\text{gen}}}) \\
&g(gen) < (s_{gen+1} - s_{\text{gen}}) \leq f(\text{gen})
\end{aligned}
\tag{3}
$$

Both rigid and soft control functions are depicted in Figure 10. This analysis was conducted exclusively with CIFAR10, revealing no significant difference in fitness scores between the naive and soft control strategies. Notably, rigid control demonstrated a slight performance advantage, as illustrated in Figure 11.

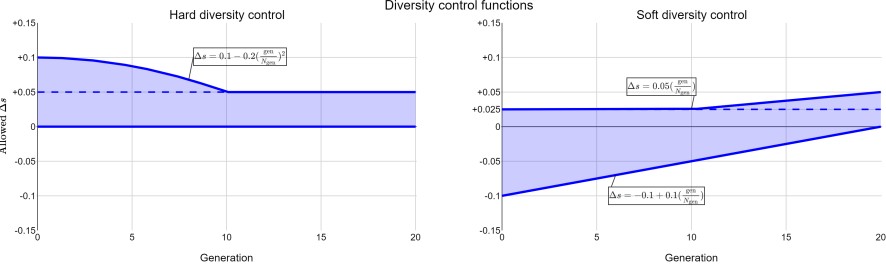

Figure 10: Potential functions proposed to regulate the allowable diversity difference between consecutive generations $\Delta s$. Rigid diversity control disallows any increase in diversity between successive generations, whereas soft diversity regulation allows for this metric to increase, facilitating a gradual shift towards exploitation while still accommodating exploration.

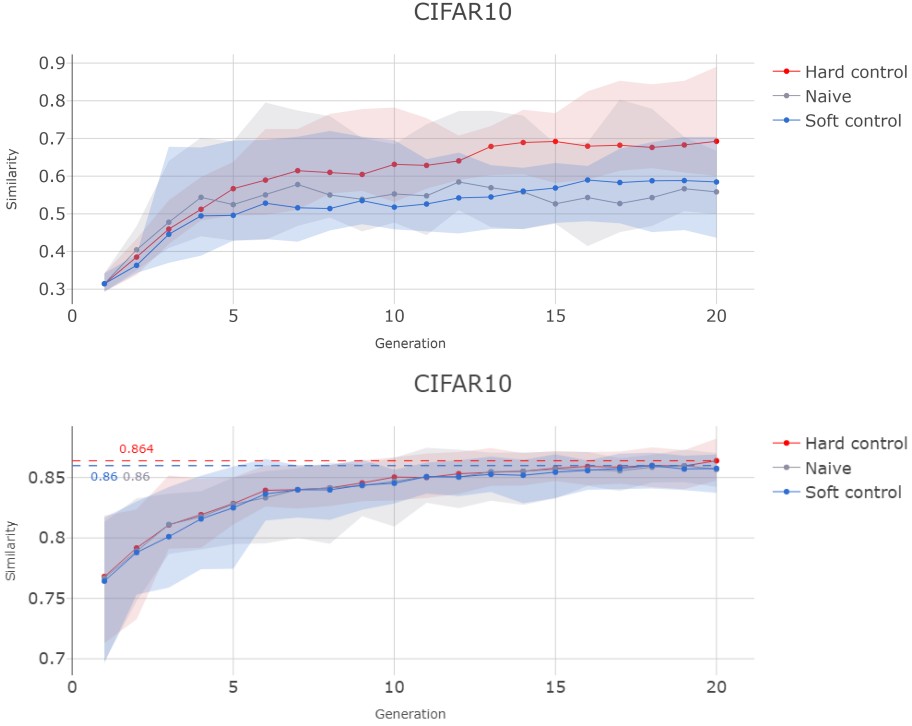

Figure 11: Comparison of similarity and fitness score evolution between Naive Evolutionary Neural Architecture Search (NAS) and our proposed approach employing Hard and Soft control functions for CIFAR10.

## B Aggregated results Across all seeds

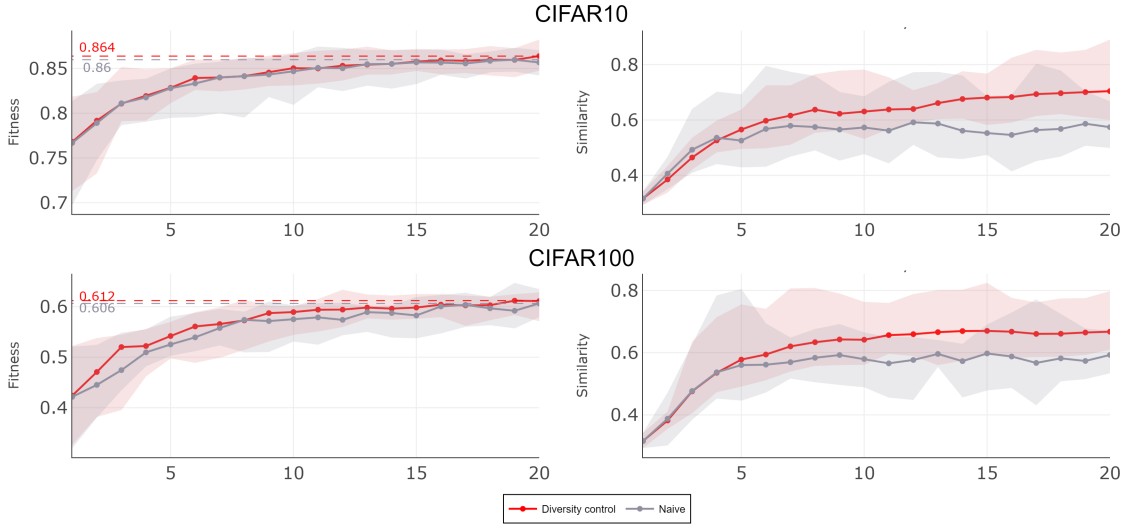

Figure 12: Evolution of Maximum Fitness Score and Population Diversity across Generations for Naive and with Diversity Control evolutionary NAS on CIFAR10 and CIFAR100. The solid line represents the median score, while the shaded area indicates the range between the maximum and minimum values observed across 10 tested seeds.

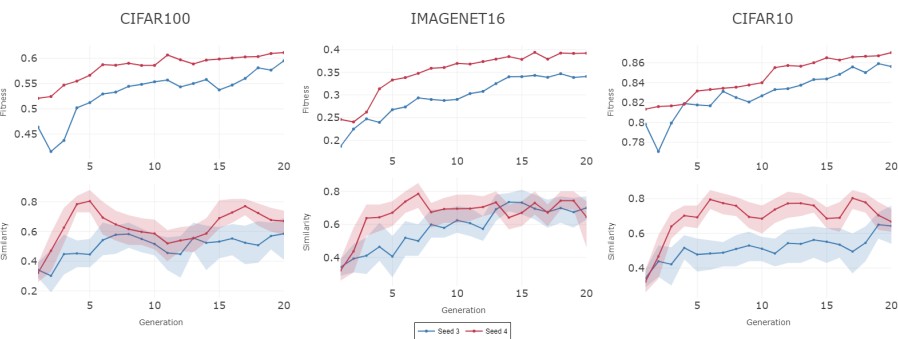

Figure 13: Max fitness and Population similarity across generations in our Naive Evolutionary Neural Architecture Search (NAS) on two initial gene pools for Imagenet16-120, CIFAR100, and CIFAR10 datasets.

## C  Results on individual seeds

To ensure the robustness of our findings and the reliability of our metric across diverse initial populations, we conducted Evolutionary NAS using 10 distinct initial seeds. The fitness accuracy and similarity, both with and without rigid diversity control, are depicted for all three datasets in Figures 14, 15 and 16. We calculate the Pearson correlation coefficient between the maximum accuracy achieved per generation and the median diversity per generation across all three tested datasets and for each of the 10 seeds. Additionally, we compute the average correlation for each case and observe that the correlation is higher when utilizing our diversity control method compared to the baseline method.

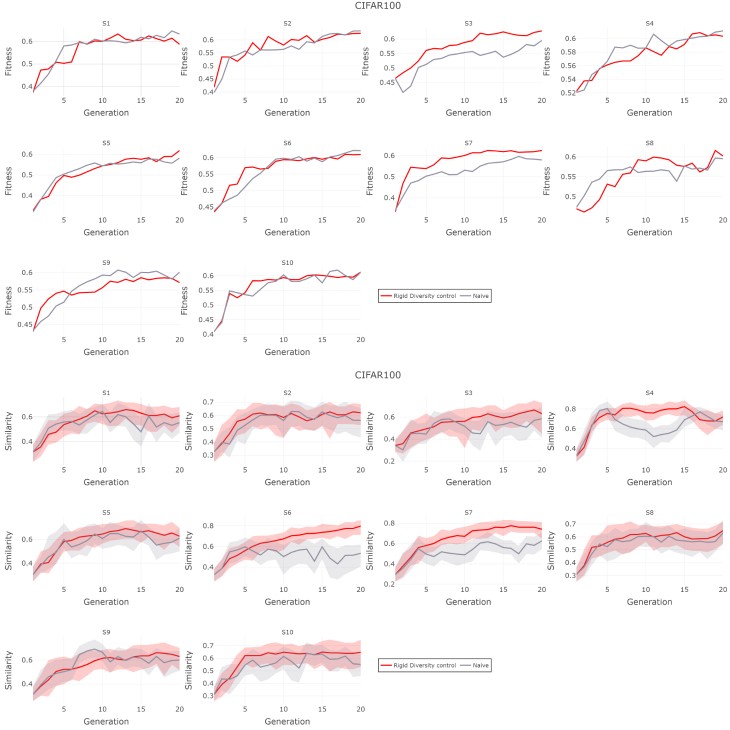

Figure 14: Maximum fitness scores and similarity across generations for each seed comparing Naive and Diversity Control Evolutionary NAS on the CIFAR-100 dataset.

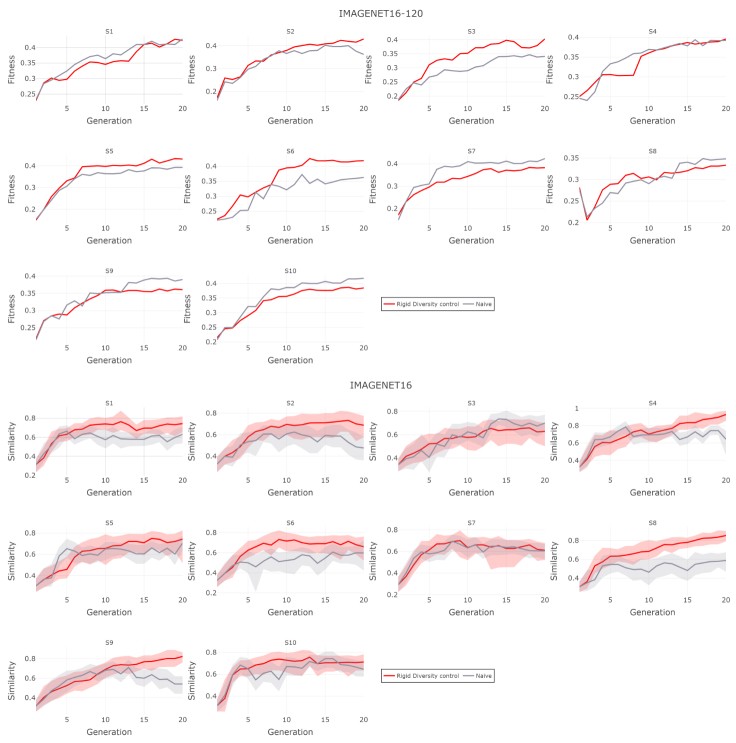

Figure 15: Maximum fitness scores and similarity across generations for each seed comparing Naive and Diversity Control Evolutionary NAS on the IMAGENET16-120 dataset.

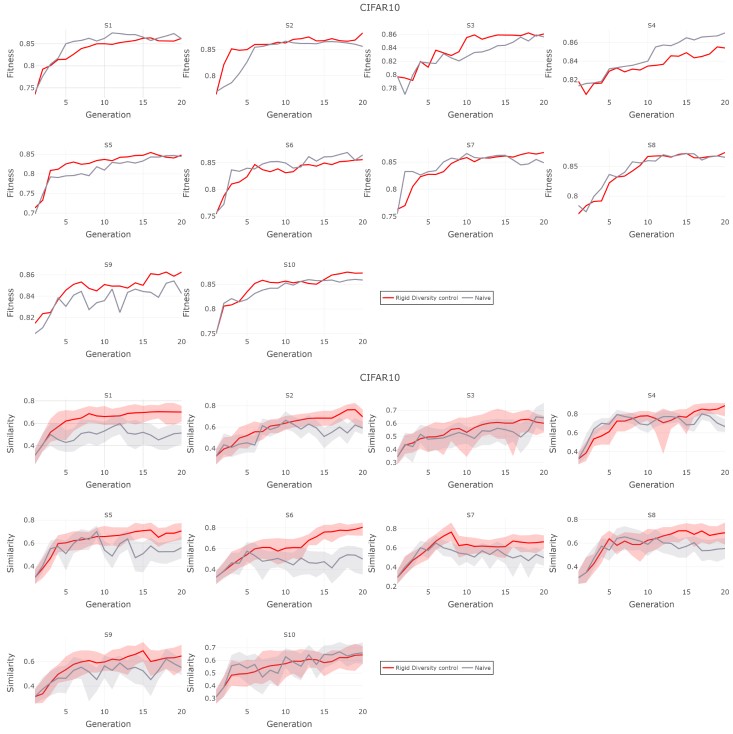

Figure 16: Maximum fitness scores and similarity across generations for each seed comparing Naive and Diversity Control Evolutionary NAS on the CIFAR10 dataset.

| seeds | ImageNet16-120 | | CIFAR10 | | CIFAR100 | |
|---|---|---|---|---|---|---|
| | *Diversity control* | *Naive* | *Diversity control* | *Naive* | *Diversity control* | *Naive* |
| 1 | 0.77 | 0.65 | 0.97 | 0.78 | 0.93 | 0.75 |
| 2 | 0.97 | 0.81 | 0.82 | 0.84 | 0.84 | 0.79 |
| 3 | 0.97 | 0.94 | 0.9 | 0.78 | 0.96 | 0.79 |
| 4 | 0.93 | 0.76 | 0.87 | 0.55 | 0.59 | 0.45 |
| 5 | 0.96 | 0.9 | 0.96 | 0.54 | 0.94 | 0.92 |
| 6 | 0.86 | 0.82 | 0.91 | 0.7 | 0.97 | 0.34 |
| 7 | 0.84 | 0.9 | 0.83 | 0.62 | 0.94 | 0.88 |
| 8 | 0.84 | 0.63 | 0.92 | 0.73 | 0.84 | 0.88 |
| 9 | 0.96 | 0.71 | 0.9 | 0.78 | 0.95 | 0.89 |
| 10 | 0.84 | 0.78 | 0.94 | 0.84 | 0.95 | 0.82 |
| **mean** | **0.894** | **0.79** | **0.902** | **0.716** | **0.891** | **0.751** |

Table 3: Pearson correlation coefficient between accuracy and median similarity for each seed when employing diversity control and the naive approach.

## D  Size and inference time of best individuals

Alongside the fitness score, we recorded the size and inference time per image of the best solutions across generations for the three datasets, as shown in Figures 17 and 18.

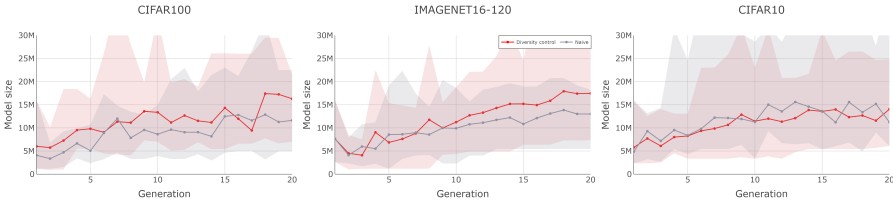

Figure 17: Comparison of sizes, in megabytes (MB), corresponding to individuals in maximum fitness scores across generations for Naive and Diversity Control Evolutionary NAS across 10 seeds. The shaded area represents the range between the maximum and minimum sizes observed among the seeds.

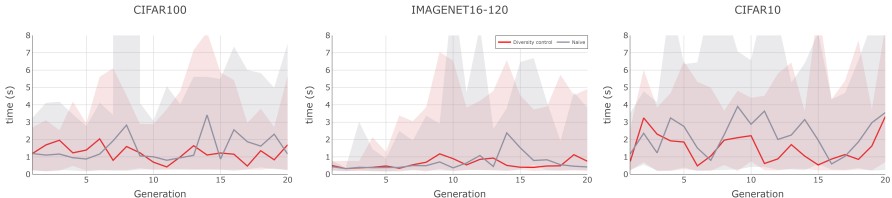

Figure 18: Comparison of inference times in seconds (s) per image, derived from individuals achieving maximum fitness scores across generations for both Naive and Diversity Control Evolutionary NAS across 10 seeds. The shaded area delineates the range between the maximum and minimum observed sizes among the seeds.

## E  Similarity of combined best individuals

We also wanted to test how similar the solutions were between the Naive strategy and the one with Diversity control. Therefore, we represented this in the clustergram shown in Figure 19.

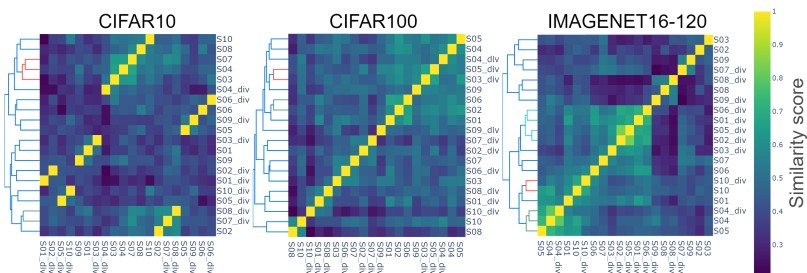

Figure 19: Hierarchical clustering based on similarity of combined Naive and Diversity-controlled best individuals, across each seed and dataset.

## F Examples of cases with significant similarity but considerable disparity in fitness scores between individuals

From the similarity matrices of each test, we selected 12 noteworthy cases and presented in Figure 20. The most frequent modifications between individuals include the deletion or insertion of a BatchNormalization(B) gene, as well as the substitution of BatchNormalization (B) with InstanceNormalization(I). In other cases, such as examples 1 and 6, where the similarity reaches values exceeding 0.97, the changes no longer occur between different types of layers or architectural structures, as they remain the same. Instead, the variations occur in the parameters associated with certain genes. In instances with more significant differences in similarity, we observe substitutions involving two or more genes, such as in example 2, where the BAB block is replaced by an InstanceNormalization (I) layer, or example 7, where the order of the D and C layers is changed.

Therefore, leveraging Figure 9 and examples from Figure 20, we can potentially narrow down or eliminate certain gene options within the search space or introduce additional rules to mitigate the generation of design patterns that consistently yield suboptimal results.

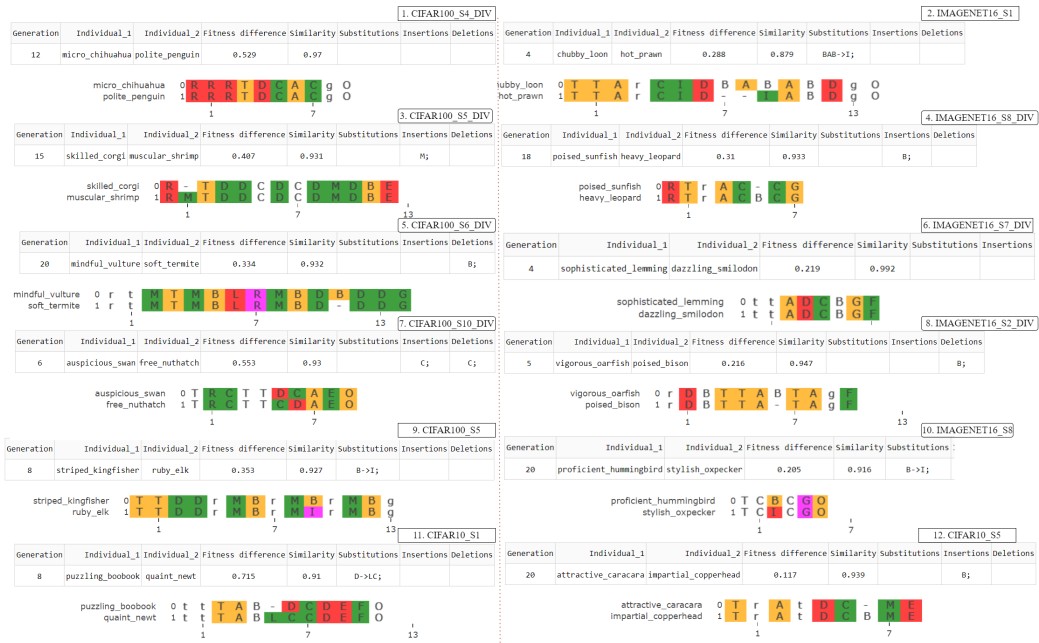

Figure 20: Alignment of Pairs of Individuals Exhibiting High Similarity but Significant Difference in Fitness Scores, Including Specific Substitutions, Insertions, and Deletions Made to the First Model (Individual_1) Resulting in Fitness Reduction in second model (Individual_2).

