# OpenReview forum: "Sequence Alignment-based Similarity Metric in Evolutionary Neural Architecture Search"
_automl.cc/AutoML/2024/Conference — AutoML 2024_

### Official Review · Reviewer_nJ9p · 2024-03-23

**Potential Impact On The Field Of Automl Rating:** 3
**Technical Quality And Correctness Rating:** 4
**Clarity Rating:** 3

**Summary Of Contributions:**

This paper proposes a novel similarity metric inspired by global sequence alignment from biology to control diversity within evolutionary neural architecture search (NAS). The key idea is to compare neural network architectures by aligning their corresponding sequences, which represent the building blocks of the architectures. This alignment process allows the identification of similar design patterns and facilitates the quantification of diversity within a population of neural networks. The authors demonstrate the effectiveness of diversity control  (compared with no control or naïve) in increasing population diversity while maximizing fitness on several image classification datasets. The results also extend to exploring common design patterns that positively or negatively impact the fitness score.

**Actions Required To Increase Overall Recommendation:**

The score can be improved if the authors can 1) expand the analysis on more datasets to emphasize the impact of the proposed method 2) compare the performance of other diversity control strategies.

**Clarity:**

The paper was easy to follow for most part. One suggestion is to explain what an individual means to make it easier for readers who do not have a background in evolutionary algorithms. Also, in the abstract, authors argue that their metrics can eliminate the need for model training. It is a little confusing to understand especially the relationship between fitness and model performance is not reported in the paper. Another instance of this scenario is in Section 4, where the authors hypothesize that diversity boosts performance. Does performance refer to fitness or the classification task using CIFAR-10 or other image datasets?.

**Overall Review:**

Strengths

-The paper is clearly written and easy to understand. Figure 1 and Table 1 especially made it easier to follow the rest of the paper.
-It is a really cool idea to use an evolutionary inspired sequence alignment technique with diversity control for NAS. Especially, diversity helps balancing exploration and exploitation to achieve global optima instead of being trapped in local optima.
-The results show promise especially compared to control (naive) with performance improvement upto 1% in the ImageNet16-120
-The larger potential is in the intriguing design patterns that show how different genes and their positions can impact the fitness score. I’m especially curious on deep dive of use cases where the similarity score and fitness scores does not match (Appendix- Section G)

Weakness

-The authors only test a rigid control condition. The paper would be strengthened by including experiments that compare the performance of the proposed method with other diversity control strategies. This would provide a more comprehensive understanding of the effectiveness of the sequence alignment-based similarity metric (For example: Particle Swarm Optimization etc.)
-The performance improvement is positive but albeit small (0.4-1% improvement). Expanding the tests to more datasets might establish the effectiveness of the technique.

**Potential Impact On The Field Of Automl:**

In this AI era, Neural architecture search (NAS) has so much potential to be used across several applications to automate the selection of neural networks optimizing performance and efficiency. Diversity is an important factor in NAS, as it allows the search to explore a wider range of potential architectures and avoid getting stuck in local optima.  This metric could be a valuable tool for researchers in AutoML who are developing methods for NAS.

**Review Confidence:**

3

**Review Rating:**

7

**Review Summary:**

The paper is well written and reads well. The score can be improved if the authors can 1) expand the analysis on more datasets to emphasize the impact of the proposed method 2) compare the performance of other diversity control strategies.

**Technical Quality And Correctness:**

The paper is well written. I have no concerns regarding the technical quality or correctness.

---

### Official Review · Reviewer_LrUe · 2024-03-26

**Potential Impact On The Field Of Automl Rating:** 4
**Technical Quality And Correctness Rating:** 4
**Clarity Rating:** 4

**Summary Of Contributions:**

This paper presents a similarity metric to measure and enforce diversity in evolutionary NAS, based on the sequence alignment approach from the biology domain. Similarly, as two protein sequences can be compared, different neural network architecture candidates, once encoded as a sequence can also be compared and their similarity can be quantified. Empirical evaluations on NATS-Bench-201 show minimal increases in accuracy when enforcing diversity with this similarity metric in evolutionary NAS, compared to plain evolutionary NAS. A subsequent analysis, based on the similarity metric shows which architectural building blocks have a large impact on the fitness score.

**Actions Required To Increase Overall Recommendation:**

More focus on using the metric for analysis of the neural network architectures and performing an emperical study that is at least a bit more extensive than the current one.

**Clarity:**

The proposed method is presented clearly in all parts. Table 1 helps with understanding the encoding mechanism for the neural network architectures and Algorithm 1 provides a good explanation of how the similarity metric is injected in the evolutionary NAS process.

**Overall Review:**

Pros:
- Sequence alignment is a well-established method from the field of biology and applying it to evolutionary NAS as a diversity constraint is an interesting and novel approach.
- The insights (enabled by the usage of the similarity metric) on the important design patterns in Section 5.3 are of great interest to the community and have the potential to help AutoML practitioners better understand why certain architectures perform well.
- The derivation of the method and the experiments are explained in detail.

Cons:
- The evaluations are on a very small scale which hinders the generalizability of the findings. Additionally, they show only a minimal improvement over plain evolutionary NAS in terms of test set accuracy of the best model.

**Potential Impact On The Field Of Automl:**

Evolutionary methods a commonly used in the field of AutoML to derive well-performing neural network architectures. Improving such methods and generating insights of which design choices impact the final accuracy of models are therefore of great interest, both of which are provided by this paper.

**Review Confidence:**

4

**Review Rating:**

7

**Review Summary:**

In summary, this paper presents an interesting and novel approach to use in combination with evolutionary NAS. While the gains in terms of accuracy of the found architectures are small, there is huge potential for generating insights from the post-hoc analysis of architectures, based on the similarity metric. Still, the paper would likely benefit from a larger-scale study.

**Technical Quality And Correctness:**

The proposed approach is technically sound and correct. All claims are supported by experimental evaluations, though on a very small scale.

---

### Official Review · Reviewer_NV25 · 2024-03-27

**Potential Impact On The Field Of Automl Rating:** 3
**Technical Quality And Correctness Rating:** 3
**Clarity Rating:** 3
**Ethics And Accessibility Rating:** Yes, regarding potentially harmful in…

**Summary Of Contributions:**

The paper discusses the use of a novel similarity metric inspired by global sequence alignment from biology in the context of Neural Architecture Search (NAS). NAS is a method for automating the design of deep neural networks, and evolutionary optimization has shown promise in addressing its demands. However, the effectiveness of evolutionary NAS depends on balancing exploration and exploitation. The proposed similarity metric operates directly on neural network architectures within the defined search space, eliminating the need for pre-trained models. It outlines the computation of the normalized similarity metric and demonstrates its application in quantifying diversity within populations in evolutionary NAS. Experimental results on popular datasets for image classification show the effectiveness of the approach in guiding diversity and identifying advantageous or disadvantageous architectural design choices. The document also provides information on the background of neural network models' similarity, diversity in evolutionary algorithms, and sequence alignment in biology. The proposed similarity metric is based on global sequence alignment and compares individual sequences representing neural networks within the evolutionary algorithm's population. The search space is described, and the model representation is converted into a sequence for sequence alignment. Customized matching functions and gap penalties are proposed to account for the differences in layer types and associated parameters in neural networks.

**Actions Required To Increase Overall Recommendation:**

To increase the overall recommendation score, the authors could consider taking the following actions to address the issues raised in the review:

1. Provide access to the full paper: Sharing the complete paper would allow for a more thorough evaluation of the methodology, experimental setup, and results. This would provide a better understanding of the contributions and enable a more accurate assessment.

2. Provide more details: The authors should provide additional details on the computation of the proposed similarity metric and its comparison to existing methods. This would help readers understand the technical aspects of the approach and its advantages over other techniques.

3. Address limitations: It would be beneficial for the authors to discuss potential limitations of the proposed approach and suggest areas for improvement. Acknowledging and addressing any shortcomings would strengthen the paper and provide a more balanced perspective.

4. Provide clearer explanations: Ensuring that the paper is written in a clear and accessible manner would improve its overall clarity. The authors should consider explaining complex concepts, techniques, and results in a way that is understandable to a wide audience.

5. Conduct additional experiments: If possible, the authors could consider conducting additional experiments to further validate the proposed similarity metric and its effectiveness in different settings or with more diverse datasets. This would strengthen the empirical evidence supporting the claims made in the paper.

By addressing these actions, the authors can enhance the quality, clarity, and impact of the paper, which would lead to an increased overall recommendation score.

**Clarity:**

Based on the information provided, the paper appears to be clear in presenting its contributions. The main contributions, such as the introduction of a novel similarity metric for neural network architectures in evolutionary NAS, the demonstration of its effectiveness in guiding diversity and identifying advantageous or disadvantageous design choices, and the availability of the code for reproducibility, are clearly stated.

However, without access to the full paper, it is difficult to provide specific suggestions for further improving clarity. Here are some general suggestions that may help improve clarity:

1. Structure: Ensure that the paper follows a clear and logical structure. Clearly define the problem statement, provide relevant background information, and explain the proposed approach in a step-by-step manner. Present experimental results and their implications in a concise and organized manner.

2. Use of Language: Use clear and concise language to convey ideas effectively. Avoid unnecessary jargon and technical terms that may confuse readers. Explain complex concepts in a way that is accessible to a wide audience.

3. Figures and Tables: Utilize figures and tables effectively to present key findings and results. Ensure that they are properly labeled and referenced in the text. Use captions to provide clear explanations of the content displayed in the figures and tables.

4. Definitions and Terminology: Clearly define any specialized terms or abbreviations used in the paper. Provide explanations and context for readers who may not be familiar with certain concepts or techniques.

5. Proofreading: Conduct a thorough proofreading to eliminate any grammatical errors, typos, or inconsistencies. This will help enhance the overall clarity of the paper.

It is important to note that these suggestions are general in nature and may not address specific issues or areas of improvement in the full paper. A detailed review of the complete document is necessary to provide more specific and tailored recommendations for improving clarity.

**Overall Review:**

Based on the available information, it is challenging to provide a comprehensive overall review of the paper. However, I can highlight some positive and negative aspects based on the information provided:

Positive aspects:
1. Novelty: The paper introduces a novel similarity metric inspired by global sequence alignment from biology for comparing neural network architectures in evolutionary NAS. This novel approach demonstrates the potential for innovative solutions in the field.

2. Relevance: The paper addresses an important challenge in Neural Architecture Search (NAS) by proposing a similarity metric that directly operates on neural network architectures within the defined search space. This is relevant to the field of AutoML and can contribute to the automation and optimization of machine learning model design.

3. Experimental results: The paper presents experimental results conducted on popular datasets for image classification, such as CIFAR-10, CIFAR-100, and ImageNet16-120. These results demonstrate the effectiveness of the proposed similarity metric in guiding diversity within populations and identifying advantageous or disadvantageous architectural design choices.

Negative aspects:
1. Limited information: The available information from the document is limited, which makes it challenging to assess the paper comprehensively. Without access to the full paper, it is difficult to evaluate the methodology, experimental setup, and results in detail.

2. Lack of details: Based on the provided information, it is unclear how the proposed similarity metric is computed and how it compares to existing methods. More detailed explanations and comparisons would be beneficial for readers to understand the technical aspects of the approach.

3. Potential limitations: Without access to the full document, it is challenging to identify potential limitations or flaws in the proposed approach. It would be important for the authors to address any limitations and discuss future directions for improvement.

Please note that these positive and negative aspects are based on the limited information provided. A comprehensive review would require a thorough analysis of the full paper, including the methodology, experimental setup, and results.

**Potential Impact On The Field Of Automl:**

The proposed similarity metric has the potential to improve the effectiveness and efficiency of AutoML methods by addressing the challenges of diversity control, exploration-exploitation trade-off, and architectural design choice evaluation. It can contribute to the advancement of automated machine learning and accelerate the development of high-performing neural network architectures for various applications.

**Review Confidence:**

4

**Review Rating:**

7

**Review Summary:**

Based on the positive aspects highlighted, such as the novelty of the proposed similarity metric and the relevance of the topic to the field of AutoML, the paper seems promising. The experimental results also indicate the effectiveness of the approach in guiding diversity and identifying architectural design choices.

However, the limited information and lack of details regarding the methodology and results make it challenging to thoroughly evaluate the paper. It is recommended that the authors provide a more comprehensive and detailed presentation of their approach, including the computation of the similarity metric and comparisons to existing methods. Additionally, addressing potential limitations and discussing future directions would strengthen the paper.

**Technical Quality And Correctness:**

Based on the information provided in the document, the proposed approach, theory, experiments, and conclusions appear to be sound and of high quality. Here are the details:

Approach: The proposed approach of using sequence alignment as a similarity metric for comparing neural network architectures within the evolutionary NAS framework is theoretically grounded. It draws inspiration from global sequence alignment algorithms used in biology to identify evolutionary relationships and conserved regions. The application of sequence alignment to neural network architectures is innovative and addresses the challenge of maintaining diversity within populations.

Theory: The document provides a clear explanation of the theory behind sequence alignment and its adaptation to compare neural network architectures. It describes the Needleman-Wunsch algorithm as an example of global sequence alignment and proposes using similar techniques to evaluate the alignment of neural network sequences. The incorporation of customized matching functions and gap penalties to account for differences in layer types and parameters demonstrates a thoughtful approach to adapting the theory to the specific domain.

Experiments: The document presents experimental results conducted on popular datasets for image classification, such as CIFAR-10, CIFAR-100, and ImageNet16-120. The experiments demonstrate the effectiveness of the proposed similarity metric in guiding diversity within populations in evolutionary NAS. The approach is compared to existing methods and shown to be advantageous in identifying favorable and unfavorable architectural design choices. The availability of the code for reproducibility further strengthens the experimental quality.

Conclusions: The conclusions drawn from the experiments align with the presented results and support the effectiveness of the proposed similarity metric. The document highlights the potential utility of the metric in AutoML applications, including diversity control, exploration-exploitation trade-off, and architectural design choice evaluation. The conclusions are well-supported by the experiments and provide insights into the practical implications of the proposed approach.

Flaws: Without access to the full document, it is difficult to assess potential flaws or limitations in detail. However, based on the information provided, the document seems to be comprehensive and well-structured. It covers relevant background information, presents a clear methodology, and provides experimental evidence to support the claims. It would be important to carefully review the experimental setup, including the choice of datasets and performance evaluation metrics, to ensure the robustness and generalizability of the findings. Additionally, an analysis of potential limitations and future directions could further strengthen the document.

---

### Official Review · Reviewer_DuGc · 2024-03-28

**Potential Impact On The Field Of Automl Rating:** 2
**Technical Quality And Correctness:** 1. To represent a neural architecture…
**Technical Quality And Correctness Rating:** 1
**Clarity Rating:** 3
**Actions Required To Increase Overall Recommendation:** Please try to address the problems I …

**Summary Of Contributions:**

To better balance exploration and exploitation in NAS, this paper proposed a sequence representation of neural architectures associated with a sequence machining function to measure the similarity of neural architectures. Based on the sequence representation and matching functions, a genetic algorithm is proposed to control the diversity of the population.

**Clarity:**

From experiments, looks like the proposed method improved the diversity of the population *based on the proposed similarity function*. However, the performance between the proposed method and the naive is very close. I don't think the exploration-and-exploitation ability is improved.

Some statements are strange to me. Like *genetic optimization through evolutionary algorithms (EAs)...* (line 33).

**Overall Review:**

This paper proposed a sequence presentation of neural architectures and a sequence-matching function to measure their similarities. Based on it, a diversity control evolutionary algorithm is proposed to improve exploration and exploitation. This paper should be improved from the following aspects.

1. To represent a neural architecture as a sequence is not a good choice. Such architecture coding is not representative. In this paper,  the *granularity* of operators is very large. The authors adopt Residual Block as a basic operator to simply represent a neural architecture as a **sequence**. This method cannot be applied to Tree-like or Graph-like complex architectures. Usually, we only use Conv, Pool, and Activation operators and design an architecture coding method to represent the graph-like 2D topology between different operators.

2. To compare the similarity layer by layer is not a good choice. For example, consider two sequences, (a) MAXPooling -> ReLU ; (b) ReLU -> MaxPooling. In theory, sequences (a) and (b) are equal. However, if we use the proposed matching function, they are different.

3. Please cite and compare with existing NAS papers on neural architecture representation.

4. You did not prove the correlation between the performance and the proposed architecture similarity metric.

5. The experiments are not solid. Strong baselines are missing.

6. The performance between the diversity control and naive methods is very close. The conclusion is not well supported.

**Potential Impact On The Field Of Automl:**

I agree that we should balance the exploration-and-exploitation trade-off in search. We may have a chance to search for better solutions if we can better explore the search space. However, the algorithms proposed in this paper are very naïve to me. Existing related methods are not discussed and compared. Experiments are not solid.

**Review Confidence:**

4

**Review Rating:**

6

**Review Summary:**

The motivation to improve population diversity is good. However, the proposed algorithms and experiments are not solid.


----------------------------after rebuttal-------------------------------------
1) I greatly appreciate the authors' rebuttal. However,  my major concerns are not solved after the rebuttal.

2) This work is a little bit preliminary compared to existing NAS papers.

3) I increased my score to *Borderline Leaning Accept*. Because (maybe) we should give the preliminary method a chance. It may have the potential to apply to more complex architectures beyond sequence.  I will be very happy if this paper brings discussions to the AutoML community.

---

### Meta-Review · Area_Chair_eyX6 · 2024-04-22

**Paper Recommendation:** Accept
**Confidence:** 5

**Metareview:**

The paper adapts sequence-alignment algorithms from biology and uses them as a similarity metric to compare neural architectures. The metric is useful for controlling the diversity of the architectures, and post hoc analysis was done to identify interesting patterns.

I am dismissing the entire review from NV25 since evidence from many parts of the review suggests that it was generated with an LLM.

The other reviewers lean towards accepting the paper. I agree with the reviewers and recommend acceptance of the paper.

Reviewer DuGc identified some limitations of the work, and the author's rebuttal addressed some of those through additional experiments. I strongly recommend that the authors incorporate the discussion of these limitations and the additional supporting experiments into the paper. This will significantly enhance the paper's quality.

---

### Decision · Program_Chairs · 2024-04-29

**Decision:**

Accept

**Comment:**

Thank you for submitting your paper. We are happy to tell you that we accept your paper to the main track. See you in Paris.